# In vivo vizualisation of mono-ADP-ribosylation by dPARP16 upon amino-acid starvation

Angelica Aguilera-Gomez[1,2], Marinke M van Oorschot[1,2], Tineke Veenendaal[3], Catherine Rabouille[1,2,3]*

[1]Hubrecht Institute-KNAW, Utrecht, The Netherlands; [2]University Medical Center Utrecht, Utrecht, Netherlands; [3]Department of Cell Biology, University Medical Center Utrecht, Utrecht, The Netherlands

**Abstract** PARP catalysed ADP-ribosylation is a post-translational modification involved in several physiological and pathological processes, including cellular stress. In order to visualise both Poly-, and Mono-, ADP-ribosylation in vivo, we engineered specific fluorescent probes. Using them, we show that amino-acid starvation triggers an unprecedented display of mono-ADP-ribosylation that governs the formation of Sec body, a recently identified stress assembly that forms in Drosophila cells. We show that dPARP16 catalytic activity is necessary and sufficient for both amino-acid starvation induced mono-ADP-ribosylation and subsequent Sec body formation and cell survival. Importantly, dPARP16 catalyses the modification of Sec16, a key Sec body component, and we show that it is a critical event for the formation of this stress assembly. Taken together our findings establish a novel example for the role of mono-ADP-ribosylation in the formation of stress assemblies, and link this modification to a metabolic stress.

**\*For correspondence:**
c.rabouille@hubrecht.eu

**Competing interests:** The authors declare that no competing interests exist.

## Introduction

ADP-ribosylation, either poly (PARylation) or mono (MARylation), is a post-translational modification that refers to the addition of one or multiple ADP-ribose units to protein substrates. It is catalysed by PARPs (also called ADP-Ribose Transferase class D, ARTD), a family of 17 proteins in mammals (*Hottiger et al., 2010*; *Leung et al., 2011*; *Leung, 2014*). PARPs have emerged as major players in several physiological processes, such as transcriptional regulation, chromatin remodelling and telomere functions (*Krishnakumar and Kraus, 2010*), cell differentiation, proliferation, apoptosis (*Hu et al., 2013*) and cellular signalling (*Watanabe et al., 2016*) as well as pathological ones, such as cancer (*Fujimori et al., 2012*) and neurodegeneration (*Cosi and Marien, 1999*).

ADP-ribosylation has also been shown to occur during cellular stress. The founding member, the nuclear PARP1, is required for DNA repair during DNA damage (*Gibson and Kraus, 2012*) where it hyper-PARylates itself as well as surrounding histones (*Gibbs-Seymour et al., 2016*). Furthermore, MARylation is linked to ER stress via PARP16, the only membrane-anchored member of this family (*Jwa and Chang, 2012*; *Di Paola et al., 2012*).

However, the progress in understanding the role of these modifications is limited by difficulties in identifying individual targets and to validate them during specific biological processes. This is mainly due to the labile bonds between ADP-ribose to the substrate, and the low abundance of this modification in steady state conditions. So far, chemical tools, such as NAD+ analogues, have been used for in vitro approaches, (*Carter-O'Connell et al., 2016*). Monitoring PARylation has been possible in vitro using the PAR affinity resin (Tulip-4301 www.tulipbiolabs.com/4301.html) and monoclonal antibodies, such as 10H and LP96-10. However, these antibodies bind poly-, not mono-ADP-ribose. This

represents a serious limitation as most PARPs are predicted to be MARylation enzymes (*Hottiger et al., 2010*; *Leung, 2014*; *Bütepage et al., 2015*). Yet, the role of this form of the modification in intracellular processes is largely unexplored.

Several biological modules, known as macrodomains, that specifically recognize either poly- or mono-ADP-ribose (*Karras et al., 2005*) (*Rack et al., 2016*) have also been used in pull down experiments for large scale proteomics studies (*Vivelo and Leung, 2015*).. Accordingly, macrodomains H2A1.1 show binding specificity for PARylated proteins (*Kustatscher et al., 2005*; *Timinszky et al., 2009*). And macrodomains from human PARP14 that exclusively binds Mono-ADP-ribose have being used to pull down MARylated PARP10 (*Forst et al., 2013*). Importantly these macrodomains do not exhibit hydrolase activity (*Rosenthal et al., 2013*; *Jankevicius et al., 2013*).

Here, we took advantage of the specificity of these macrodomains to design, build and fine-tune stable MARylation (MAD) and PARylation (PAD) detection probes. We then used them to follow PARylation and MARylation in vivo during cellular stresses, with particular focus on amino-acid starvation that induces the formation of a recently described stress assembly, the Sec body (*Zacharogianni et al., 2014*).

Sec body formation results from the inhibition of a major anabolic pathway, the protein transport through the secretory pathway, upon amino-acid starvation of Drosophila cells (*Amodio et al., 2009*; *Zacharogianni et al., 2014*). The secretory pathway ensures the delivery of signal peptide containing proteins to the extracellular medium and the plasma membrane and to nearly all membrane-bound compartments. After the synthesis in the endoplasmic reticulum (ER), they exit this organelle in COPII coated vesicles that form and bud at defined sites on the ER, called ER exit sites (ERES). COPII formation requires Sar1, its GEF Sec12 and the structural proteins forming the coat itself, Sec23/24 and Sec13/31 (*Miller and Schekman, 2013*) as well as the large hydrophilic scaffold protein Sec16 (*Sprangers and Rabouille, 2015*). Newly synthesized proteins then reach the Golgi apparatus where they are processed, sorted and dispatch to their final destination.

The inhibition of protein transport through the secretory pathway upon amino-acid starvation is accompanied by the remodeling of ERES and the formation of a novel type of pro-survival stress assembly with liquid droplet properties, the Sec body, where COPII coat proteins and Sec16 are stored and protected from degradation during the period of stress (*Zacharogianni et al., 2014*).

Using YFP-PAD, we show that PARylation is not prominent during amino-acid starvation. Conversely, using MAD, we show that MARylation is strongly induced by this nutrient stress. Furthermore, we demonstrate that this modification is required for the formation of Sec bodies. We identify dPARP16 as the enzyme necessary and sufficient to catalyse MARylation and Sec body formation during amino-acid starvation. Last, we identify the ERES component Sec16 as a novel dPARP16 substrate and show that it is MARylated on its C-terminus in an amino-acid starvation specific manner. We propose that this event initiates the formation of the Sec bodies and poses Sec16 as a stress response protein.

Taken together, our findings establish an unprecedented example for the role of mono-ADP-ribosylation in the formation of stress assemblies, and link this modification to a metabolic stress. Furthermore, this demonstrates that the macrodomain-based probes that we built are useful and specific tools to follow ADP-ribosylation taking place during biological processes in vitro and in vivo. We propose that the visualization of ADP-ribosylation will shed light on PARPs function during specific biological processes and illustrates the physiological relevance of these post-translational modifications during stress. In this regard, we have identified dPARP16 as a novel key factor in cell survival to amino-acid starvation.

## Results

### Visualising PARylation and MARylation in vivo upon cellular stress

To visualise whether PARylation events take place during amino-acid starvation, we engineered a PARylation detection probe (PAD) using the human macrodomain H2A1.1 that specifically recognises poly-ADP-ribose (*Kustatscher et al., 2005*; *Timinszky et al., 2009*; *Forst et al., 2013*) fused to YFP (YFP-PAD) (*Figure 1A*). When expressed in Drosophila S2 cells, YFP-PAD is both cytoplasmic and nuclear (*Figure 1B*). Amino-acid starvation only elicits a weak remodelling of the probe in the cytoplasm and an increase of its nuclear pool (*Figure 1B*). To validate the functionality of the probe,

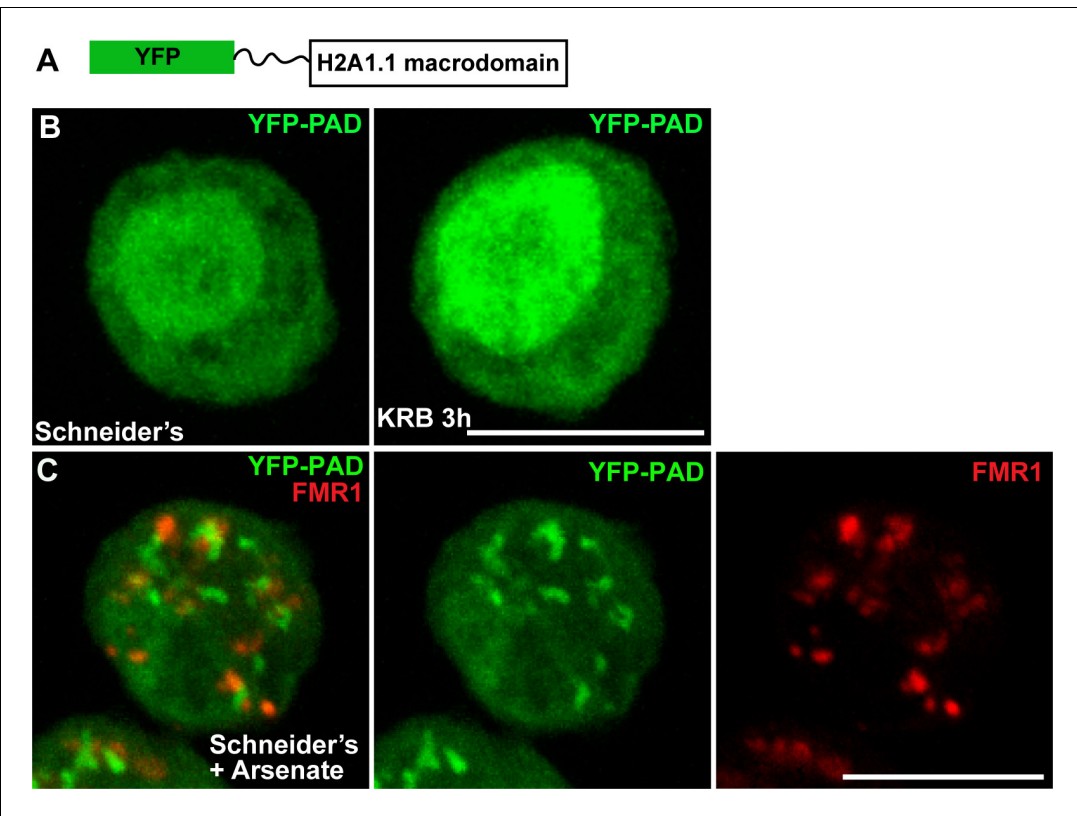

**Figure 1.** PAD in cellular stress. (**A**) Schematics of YFP-PAD probe. (**B**) YFP-PAD in growing (Schneider's) and amino-acid starved (KRB) cells for 3 hr. Note that with the exception of an increase of the nuclear intensity, amino-acid starvation does not lead to the formation of a cytoplasmic pattern. (**C**) YFP-PAD in S2 cells upon arsenate treatment. Note the formation of a robust YFP-PAD cytoplasmic pattern that co-localises with stress granules (FMR1, red). Scale bars: 10 μm

The following source data is available for figure 1:

**Source data 1.** List of the primers used in this manuscript.

we treated S2 cells with arsenate, a treatment that, in human cells, elicits PARylation known to be required for stress granules integrity (*Leung et al., 2011*). Accordingly, YFP-PAD forms defined cytoplasmic structures in arsenate treated S2 cells (*Figure 1C*) that partially co-localise with stress granules (marked by FMR1, *Figure 1C*), showing that YFP-PAD is functional. Taken together, these results indicate that amino-acid starvation does not elicit detectable PARylation events.

We then asked whether amino-acid starvation elicits MARylation. To approach this, we engineered an optimised a MARylation detection (MAD) probe based upon the macrodomains 1–3 of human PARP14. Indeed, crystallography of these macrodomains have revealed a conserved fold that binds Mono-ADP-ribose. Furthermore, calorimetric affinity assays show that the affinity for mono-ADP-ribose is contributed by the three macrodomains in a cooperative manner, whereas each macrodomain taken individually does not bind the moiety. Last, hPARP14 macrodomains 1–3 bind specifically MARylated, but not PARylated, substrates in vitro (*Forst et al., 2013*). Therefore, the PARP14 macrodomains 1–3 were GFP-tagged at their N-terminus, and a linker was inserted to preserve their binding capabilities (*Figure 2A*, *Figure 2—figure supplement 1*).

When expressed in S2 cells under growing conditions, GFP-MAD is diffuse in the cytoplasm in most of the transfected cells, but in contrast to YFP-PAD is absent from the nucleus (*Figure 2B,B'*). When cells are starved of amino-acids for increasing length of time, GFP-MAD adopts a defined pattern. After 1 hr, it concentrates in one spot in a low percentage of cells (arrows in *Figure 2B,B'*). After 2 hr, it forms 1–3 spots in a larger number of cells. Thereafter, the number of spots (some of

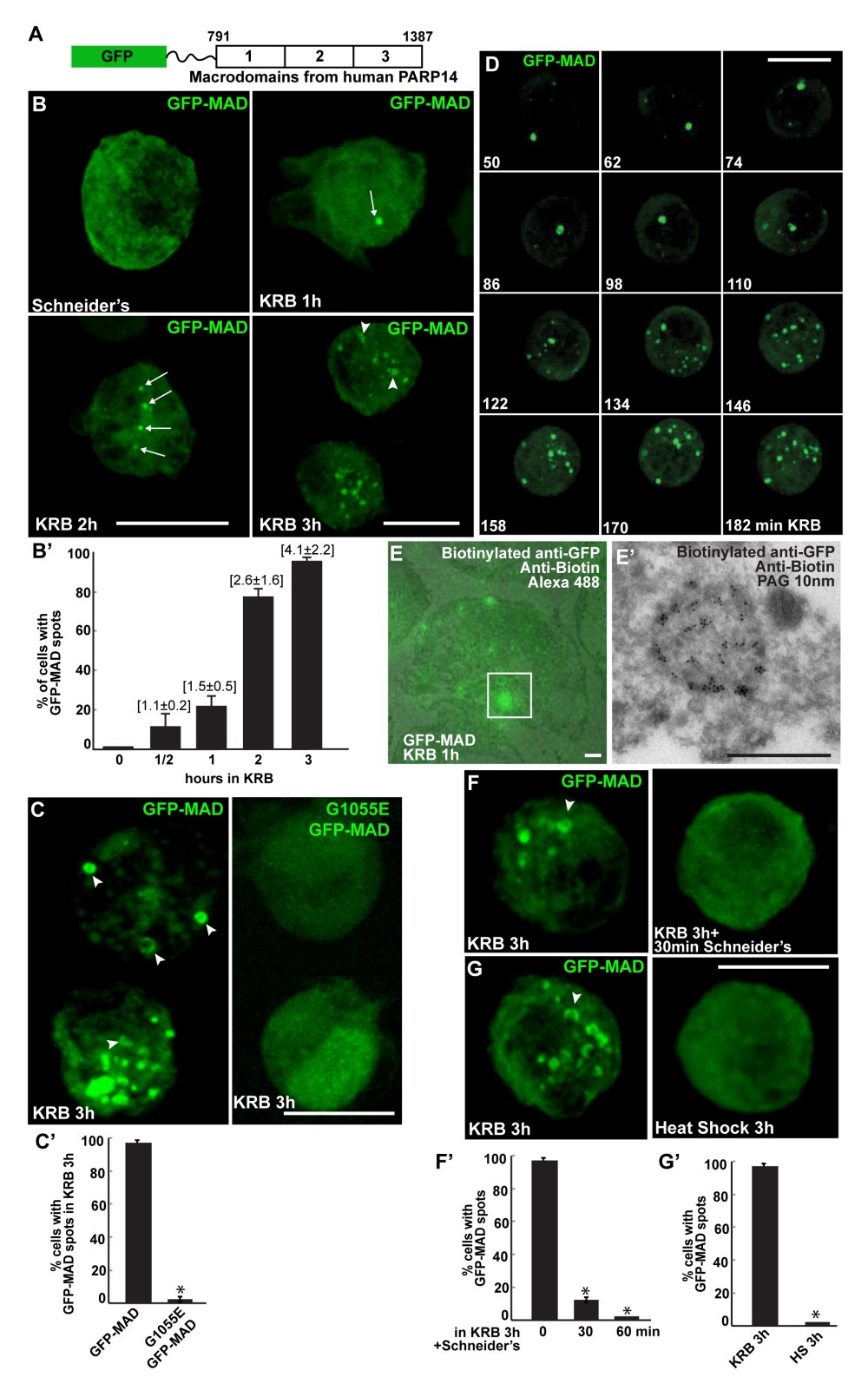

**Figure 2.** Amino-acid starvation leads to the formation of MARylation spots visualised with GFP-MAD. (**A**) Schematics of the GFP-MAD probe. (**B, B'**) Fluorescence of GFP-MAD in growing S2 cells (Schneider's) and upon amino-acid starvation (KRB) for increasing amount of time as indicated (**B**). Note the formation of GFP-MAD spots (arrows) (some in an U-shape, arrowheads in B-D). The % of cells at each time point displaying GFP-MAD spots is shown in **B'**. The average number of spots per cell is indicated above each bar. (**C, C'**) Fluorescence of GFP-MAD and G1055E GFP-MAD probe (that

*Figure 2 continued on next page*

*Figure 2 continued*

does not bind mono-ADP-ribose in vitro). Note that the mutant probe does not form spots in KRB (quantified in **C'**). (**D**) Stills of a time-lapse movie (*Video 1*) of GFP-MAD in cells incubated in KRB for 3 hr. The first frame is taken after 50 min incubation. The subsequent frames are taken every 12 min. (**E, E'**) Correlative Fluorescence/IEM of GFP-MAD spots in S2 cells upon amino-acid starvation (KRB, 1 hr). The IEM (**E'**) corresponds to the white rectangle in fluorescence that is overlapped with the corresponding electron micrograph (**E**). (**F, F'**) Fluorescence pattern of GFP-MAD and endogenous Sec16 (red) in KRB and in KRB followed by incubation with Schneider's for 30 min. Note that GFP-MAD pattern is completely reverted (quantified in **F'**). (**G, G'**) Fluorescence pattern of GFP-MAD in KRB and upon heat shock (3 hr at 37°C). Scale bars: 10 μm (**B, C, D, F, G**); 500 nm (**E, E'**). Error bars: SEM.

The following figure supplement is available for figure 2:

**Figure supplement 1.** GFP-MAD design and optimisation.

them with a U/donut shape) (arrowheads, *Figure 2B–E*) increases and they tend to concentrate in the middle of the cell after 4 hr (not shown). These results suggest that amino-acid starvation triggers cytoplasmic MARylation events that are visualised using GFP-MAD.

To demonstrate that the GFP-MAD spots forming upon amino-acid starvation correspond to the detection of MARylation events, a point mutation (G1055E) was introduced in the macrodomain two of GFP-MAD (*Figure 2A*). This mutation affects the ADP-ribose binding pocket and therefore interferes with the ADP-binding activity (*Forst et al., 2013*) and in vitro to completely abrogate its binding to mono-ADP-ribose (*Dani et al., 2009*; *Karras et al., 2005*). Strikingly, amino-acid starvation of S2 cells expressing G1055E GFP-MAD does not result in any pattern and the mutant probe remains diffuse in the cytoplasm (*Figure 2C,C'*).

Next, we visualised MARylation events in live cells. This confirms that MAD spots start forming about 1 hr after starvation and accumulate. It also shows that they are overall stable, (*Figure 2D* and *Video 1*), although few appear more transient with an average lifetime of 50 min (not shown). Using a Fluorescence-to-ImmunoEM correlative method (*Vicidomini et al., 2010*; *Hassink et al., 2012*), we showed that GFP-MAD spots correspond to non-membrane bound structures (*Figure 2E,E'*), ranging from 600 nm to two microns in diameter. We tested their reversibility upon nutrient replenishment following starvation and found that they are fully reversible after 30 min of Schneider's addition (*Figure 2F,F'*). Last, we show that heat stress (*Figure 2G,G'*) and arsenate treatment (not shown) do not elicit a GFP-MAD pattern, confirming the specificity of MARylation to amino-acid starvation.

Taken together, we demonstrate that GFP-MAD detects localised MARylation events that are specifically triggered by amino-acid starvation.

## dPARP16 activity controls MARylation events upon amino-acid starvation

In order to identify whether and which PARPs are involved the amino-acid starvation driven MARylation events, we searched for Drosophila PARPs using psi BLAST and HHpred with the canonical dPARP1/CG40441 as query. In line with (*Hottiger et al., 2010*), we identified three additional ORFs: CG4719 is homologous to human Tankyrase, CG15925 is homologous to human PARP16, and CG18812 is homologous to human GDAP2, a macrodomain, not a PARP (*Rack et al., 2016*). It was therefore not considered here.

**Video 1.** GFP-MAD time-lapse movie of one cells incubated in KRB (t = 0) for 3 hr. One frame was taken every 10 min and the movies are displayed at 12 frame/s (related to *Figure 2D*).

We tested these PARPs for their role in GFP-MAD spot formation upon amino-acid starvation and showed that MAD spot formation strictly depends on dPARP16. First, dPARP16 depletion completely prevents their formation upon amino-acid starvation and GFP-MAD remains diffuse in the cytoplasm (*Figure 3A,A'*). In comparison, the depletion of the other PARPs has no effect and GFP-MAD spots form as in mock-depleted cells (*Figure 3—figure supplement 1*). Second, the over-expression of dPARP16 under growing conditions induces the robust formation of GFP-MAD spots in most of the cells (*Figure 3B,B',E*). Importantly, 84 ± 6% of GFP-MAD spots formed upon dPARP16 overexpression partially or completely co-localize with the enzyme (*Figure 3B,B'*). These results suggest that dPARP16 mediates the MARylation response to amino-acid starvation.

dPARP16 is the closest homolog of human PARP16 as shown by building a phylogenetic tree containing dPARP16 and all human PARPs (*Figure 3—figure supplement 2*). Both enzymes are of similar length and they have similar catalytic domain. Human PARP16 has a catalytic site comprising the triad HYY (*Hottiger et al., 2010*) and using HHpred, we predicted that the catalytic site of dPARP16 consists of the YYY triad (Y199, Y221, Y284) (*Figure 3—figure supplement 3*). Of note, flies are the outliers as PARP16 in C.elegans and Xenopus do have a HYY site. Last, as hPARP16, dPARP16 is also predicted to be membrane anchored with catalytic domain facing the cytoplasm (*Jwa and Chang, 2012*; *Di Paola et al., 2012*) (*Figure 3C*). Accordingly, we find that dPARP16 localises to ER (*Figure 3—figure supplement 4*). In support of this localisation, overexpression of dPARP16 remodels the ER (as shown with KDEL receptor and calnexin) in a very similar fashion as amino-acid starvation does *Figure 3—figure supplement 5*). This phenotype strengthens the notion that dPARP16 is localized at the ER and that the ER remodelling is a result of its activation.

To address the role of dPARP16 catalytic activity in GFP-MAD detected MARylation, we generated the dPARP16 catalytic mutant Y221A. We showed that its expression does not lead to the formation of GFP-MAD spot (*Figure 3D,F*), whereas expression of the wild type dPARP16 does (*Figure 3B,B',F*). This indicates that the integrity of dPARP16 catalytic site is required for its MARylation activity.

We then addressed the role of membrane anchoring in dPARP16 function by expressing the dPARP16 cytoplasmic domain (ΔTM dPARP16). Unlike the wild type protein (*Figure 3B',F*), expression of the ΔTM dPARP16 does not induce the formation of GFP-MAD spots (arrowheads in *Figure 3E,F*), indicating that dPARP16 membrane anchoring is required for its MARylation activity.

Last, we show expression of dPARP16 does not elicit a PARylation pattern using YFP-PAD (*Figure 3G,G'*).

Taken together, these results show that upon amino-acid starvation, ER membrane-bound dPARP16 catalyses localised MARylation events that are detected by GFP-MAD. This makes GFP-MAD a sensor detecting dPARP16 catalytic activity in vivo.

## dPARP16 is necessary and sufficient for Sec body formation

We have recently shown that amino-acid starvation drives the formation of a novel stress assembly related to the early secretory pathway, the Sec body. Sec bodies are pro-survival cytoplasmic stress assembly that incorporate and protect COPII subunits and ERES components from degradation (*Zacharogianni et al., 2014*) (*Figure 4A'*). By immunofluorescence, Sec bodies appear as bright circular structures of 700 ± 100 nm in diameter (confirmed by immuno-electron microscopy, IEM) and there are typically 7 ± 3 Sec bodies per starved cell (*Zacharogianni et al., 2014*). They are distinct from ERES that are more numerous (about 15 ± 7), appear fainter and have a more irregular shape (*Figure 4A,A'*).

We assessed whether amino-acid starvation driven dPARP16 dependent MARylation events are linked to Sec body formation by testing the role of this enzyme in their formation. We found that dPARP16 depletion completely prevents Sec body formation upon amino-acid starvation (*Figure 4B, B'*). We had shown that inhibition of Sec body formation strongly affects cell survival (*Zacharogianni et al., 2014*). In agreement with this, dPARP16 depletion also strongly affects cell survival upon amino-acid starvation as well as cell recovery upon stress relief (*Figure 4C*). In contrast, in dPARP16 depleted cells during growing conditions cell survival is unaffected, indicating that dPARP16 plays a crucial role exclusively upon starvation. These results together demonstrated that dPARP16 is a key factor in the survival response to amino-acid starvation

Furthermore, expression of dPARP16 quantitatively drives the specific formation of Sec bodies under growing conditions (*Figure 4D*, arrows, *Figure 4D'*). This was confirmed by immuno-EM

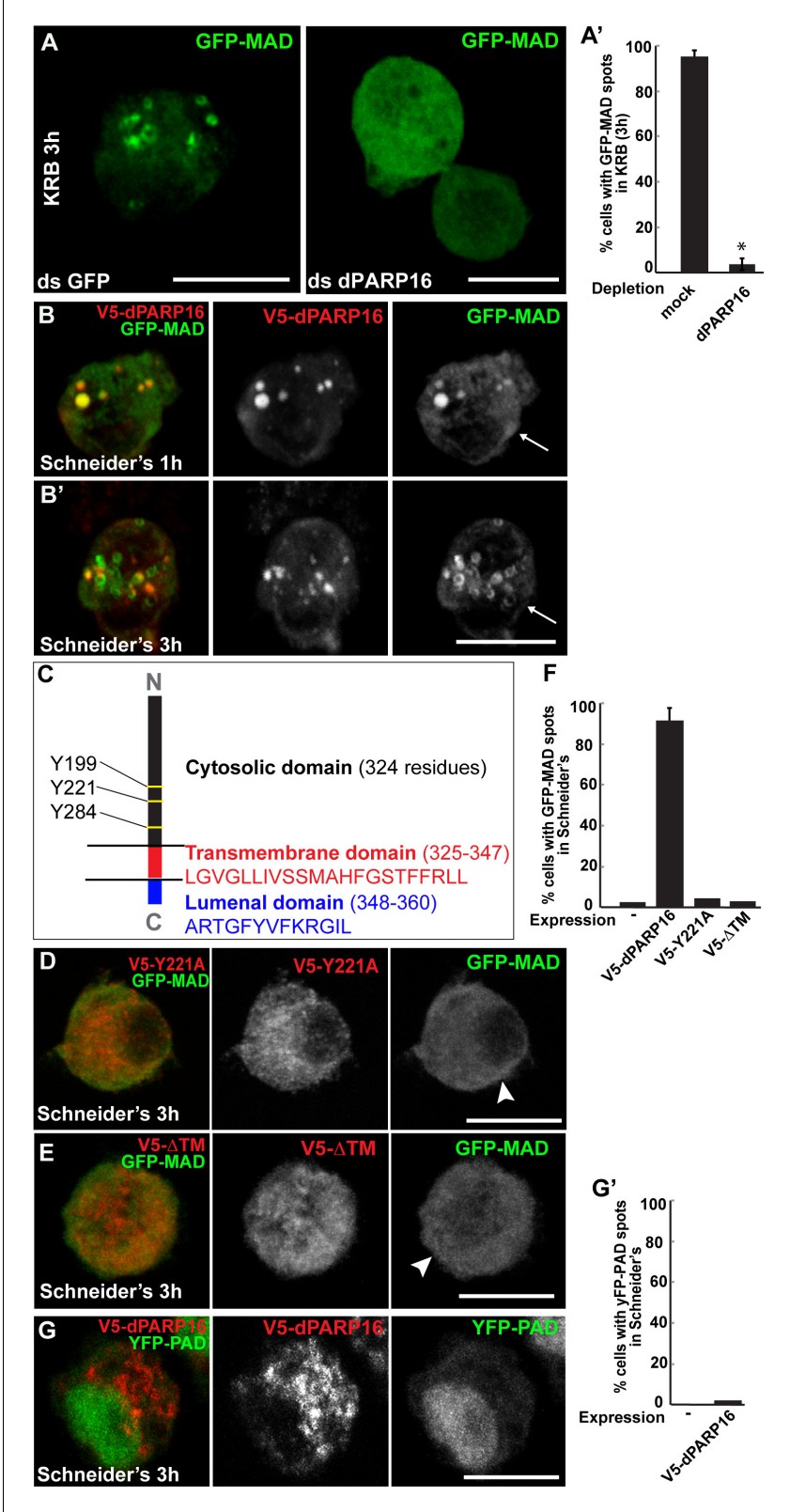

**Figure 3.** Amino-acid starvation triggered MARylation events are dPARP16 dependent. (**A, A'**) Visualisation of GFP-MAD in mock and dPARP16 depleted cells upon amino-acid starvation (KRB) (**A**). Note that dPARP16 depleted cells do not exhibit GFP-MAD spots (quantified in **A'**). (**B, B'**) Visualisation of GFP-MAD in S2 cells expressing V5-dPARP16 in growing conditions. Note that the enzyme expression drives the formation of GFP-MAD

*Figure 3 continued on next page*

*Figure 3 continued*

spots in the absence of stress and that GFP-MAD spots strongly co-localise with the enzyme (quantified in F). (**C**) dPARP16 has 359 residues including a transmembrane domain of 22 (in red) and a luminal domain of 12 (in blue). The TM has been predicted using the TMHMM server V.2.0 (http://www.cbs.dtu.dk/services/TMHMM-2.0/). The three tyrosines making up the catalytic sites are marked. (**D–F**) Visualisation of GFP-MAD in S2 cells expressing Y221A V5-dPARP16 catalytic mutant (**E**) and ΔTM V5-dPARP16 (**F**) in growing conditions. Note that none of the mutated forms of dPARP16 elicits GFP-MAD spot formation (arrowheads) (quantified in F). (**G, G'**) Visualisation of YFP-PAD in S2 cells expressing V5-dPARP16 in growing conditions. Note that the YFP-PAD localisation does not change (quantified in **G'**). Scale bars: 10 μm. Error bars: SEM

The following figure supplements are available for figure 3:

**Figure supplement 1.** Screen for PARPs in MAD spot formation.

**Figure supplement 2.** Drosophila PARP16 is the homologue of human PARP16.

**Figure supplement 3.** Comparison between Drosophila and human PARP16 catalytic site.

**Figure supplement 4.** dPARP16 is anchored at the ER.

**Figure supplement 5.** dPARP16 is anchored at the ER.

(*Figure 4E*). dPARP16 is therefore necessary and sufficient for Sec body formation as it is for GFP-MAD spot formation. Accordingly, depletion (*Figure 4—figure supplement 1*) and overexpression (*Figure 4—figure supplement 2*) of dPARP1 and dTNK do not affect Sec body formation.

We further tested whether dPARP16 catalytic activity is required for Sec body formation. Whereas expression of wild type PARP16 leads to Sec body formation, the expression of dPARP16 catalytic mutants (Y199A and Y221A) do not (*Figure 5A,A'*). To confirm this result, we show that the expression of Y199A dPARP16 does not rescue Sec body formation in starved dPARP16 depleted cells, whereas the expression of wild type dPARP16 does (*Figure 5B,B'*). We also tested the role of membrane anchoring in dPARP16 function in Sec body formation by expressing the dPARP16 cytoplasmic domain (ΔTM dPARP16). We show that unlike the wild type protein, it does not induce Sec body formation (*Figure 5C,A'*, arrowheads).

These results show that dPARP16 coordinates the response to amino-acid starvation, i.e. the MARylation events and Sec body formation. In support of this, we found that dPARP16 localises near or around Sec bodies (*Figure 5C,D*, small green arrows), suggesting that Sec body components could be dPARP16 MARylated substrates.

## Sec16 is MARylated upon amino-acid starvation

To address whether Sec body components are dPARP16 substrates, we first visualised GFP-MAD kinetics during amino-acid starvation with respect to Sec body formation. We found that a large proportion of GFP-MAD spots that form after 1-2 hr incubation in KRB overlap with, or are in close proximity to, ERES and small forming Sec bodies (*Figure 6A,A'*), suggesting that ERES components could be MARylated prior to Sec body formation.

Furthermore, we found that a significant number of Sec bodies (up to 40% after 3 hr incubation in KRB) are formed adjacent to, or overlapping with, GFP-MAD spots (*Figure 6B,B'*, arrowheads). In agreement, a small but consistent pool of Sec16 is observed within the GFP-MAD spots (*Figure 6C, D*). We also found a small pool of GFP-MAD within Sec bodies (*Figure 6D'*), but overall GFP-MAD presence within Sec bodies is weak. Together, this suggests that MARylation of ERES components is linked to Sec body formation.

To strengthen the notion that Sec body components are MARylated upon amino-acid starvation, we focused on Sec16, a key Sec body component (*Zacharogianni et al., 2014*). We set up an in vivo MARylation assay as an alternative to the classical in vitro one that uses purified components. To do this, we designed an anchoring-away strategy where by Sec16 is tagged at its C-terminus with the CAAX motif of Ras that efficiently anchors it to the plasma membrane (*Hancock et al., 1991*), and is

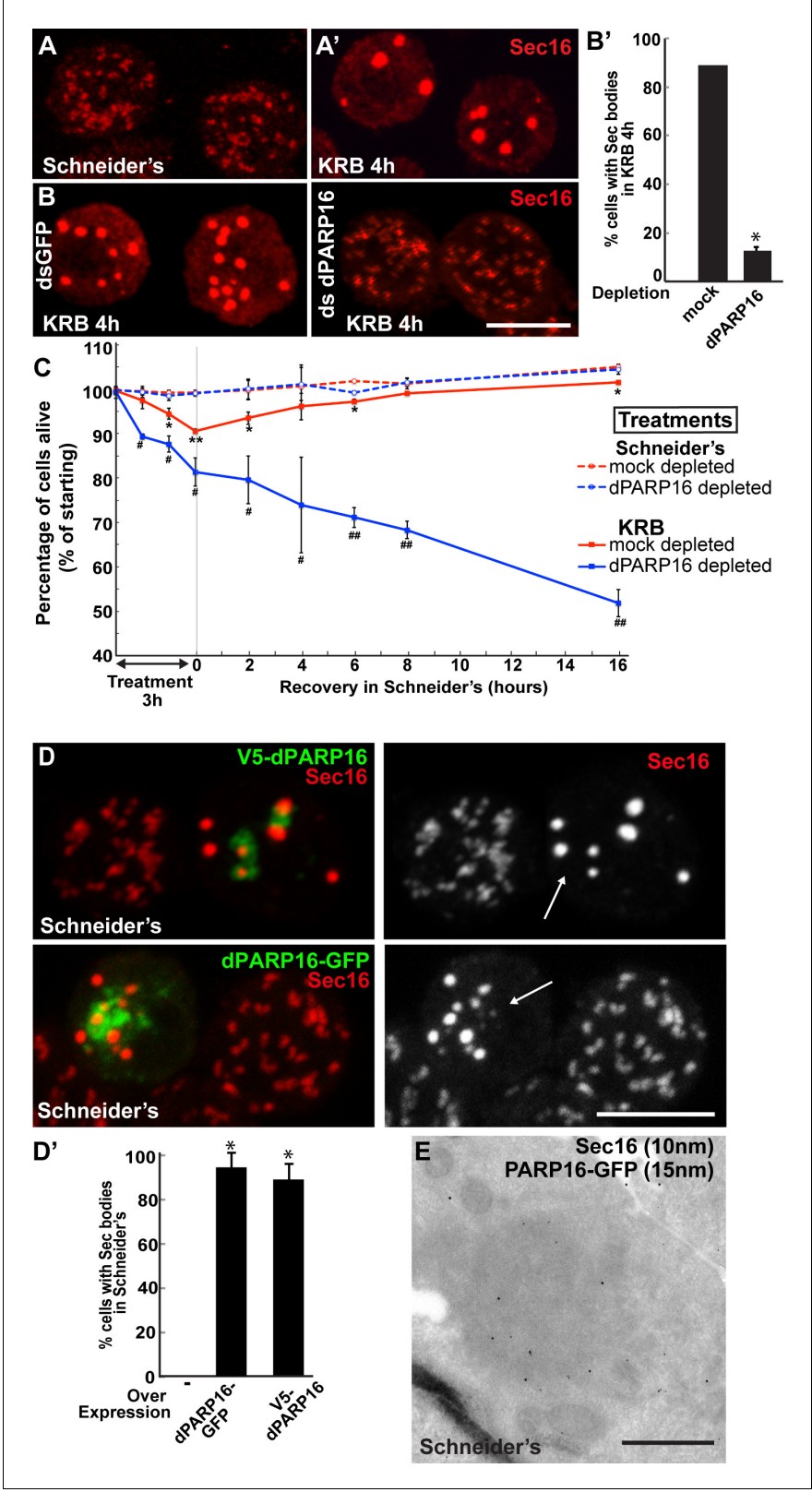

**Figure 4.** dPARP16 is required for amino-acid starvation driven Sec body formation. (**A**) Immunofluorescence (IF) visualisation of endogenous Sec16 (red) at the ERES in growing Drosophila S2 cells (Schneider's) and in Sec bodies upon amino-acid starvation (KRB). (**B–B'**) IF visualisation of endogenous Sec16 (red) in mock and dPARP16 depleted S2 cells upon amino-acid starvation (KRB) (**B**). Note that dPARP16 depletion inhibits Sec body formation

*Figure 4 continued on next page*

*Figure 4 continued*

(quantified in **B'**). (**C**) Graph of cell viability (expressed as percentage of alive cells) upon 'treatments' as indicated and recovery. The number of starting cells at t = 0, mock- (dsGFP, red lines) and dPARP16 depleted (blue lines) is set at 100%. These cells are incubated in Schneider's (dashed lines) and KRB (solid lines) for 3 hr followed by further incubation in Schneider's for 16 hr. Note that the dPARP16 depleted cells are more sensitive to starvation than the controls and they do not recover, whereas their viability not affected when grown in full medium. p-values were calculated for each time point corresponding to mock-depleted cells incubated in Schneider's and in KRB. * marks p-values higher the $10^{-2}$ and **p-values higher than $10^{-5}$. p-values were also calculated for each time point corresponding to mock and dPARP16 depleted cells incubated in KRB. # marks p-values higher the $10^{-2}$ and ##, p-values higher than $10^{-4}$. (**D–D'**) IF visualisation of endogenous Sec16 (red) in cells) over-expressing dPARP16-GFP and V5-dPARP16 in growing cells (Schneider's) (**C**). Note that it drives the robust formation of Sec bodies (arrows in **C**) (quantified in **D'**). (**E**) Immuno-electron microscopy (IEM) visualisation of endogenous Sec16 (10 nm gold) in dPARP16-GFP overexpressing cells (15 nm). Scale bars: 10 µm (**A, B, D**); 500 nm (**E**). Error bars: SEM (**A, D'**) and SD (**C**).

The following figure supplements are available for figure 4:

**Figure supplement 1.** dPARP1 and dTNK depletion does not affect Sec body formation upon amino-acid starvation.

**Figure supplement 2.** dPARP1 and dTNK overexpression does not lead to Sec body formation in growing conditions.

co-expressed with cherry-MAD. Given that this probe specifically binds MARylated substrates (*Figure 2C*), the reasoning is that if Sec16-CAAX is MARylated upon amino-acid starvation, it will recruit cherry-MAD to the plasma membrane.

First, we verified that Sec16-GFP-CAAX expression results in its localisation to the plasma membrane. From there, it is able to recruit other Sec body components specifically upon amino-acid starvation (such as Sec23, *Figure 7—figure supplement 1*, arrows). As a result, Sec bodies are no longer formed in the cytoplasm (*Figure 7—figure supplement 1*). When cherry-MAD and Sec16-GFP-CAAX are co-transfected in S2 cells in growing conditions, the cherry-MAD remains diffuse (*Figure 7A*). However, it strongly co-localises with Sec16-GFP-CAAX at the plasma membrane upon amino-acid starvation (*Figure 7A',E*). These results suggest that Sec16 is likely MARylated. Interestingly, cherry-MAD is not recruited to the plasma membrane by Sec23-GFP-CAAX (*Figure 7B,E*) upon amino-acid starvation, suggesting substrate specificity.

To confirm this result, we used GFP-TRAP to immuno-precipitate GFP-MAD from cell lysates prepared from growing and amino-acid starved S2 cells. Cells expressing GFP were used as control. GFP-MAD binds Sec16 significantly more upon amino-acid starvation than under growing conditions and more than GFP (*Figure 7C,D*). To further test the specificity of the GFP-MAD/Sec16 interaction, we investigated whether Sec23 (*Figure 7D*) and RNA binding proteins (FMR1, Caprin and Rasputin) (*Figure 7C* and not shown) are also pulled down by GFP-MAD upon amino-acid starvation. We found that neither of them are. These results show that Sec16 is likely MARylated upon amino-acid starvation, although we cannot rule out that Sec16 could also be bound to a MARylated substrate that is recognized by GFP-MAD.

## dPARP16 dependent Sec16-SRDC MARylation is a key event in Sec body formation

To identify the region of Sec16 that is recognized by GFP-MAD upon amino-acid starvation (and that is MARylated), we employed the same CAAX anchoring-away strategy as above on Sec16 truncations. Removing the N-terminus of Sec16 (ΔNC1-GFP-CAAX) does not alter cherry-MAD recruitment to the plasma membrane upon starvation (*Figure 7E*). In contrast, truncation of Sec16 C-terminus (ΔCter-GFP-CAAX) completely abolishes cherry-MAD recruitment to the plasma membrane upon starvation (*Figure 7E*). Instead, cherry-MAD form spots in the cytoplasm (red arrows *Figure 7—figure supplement 2A*). Accordingly, co-expression of Cter-GFP-CAAX with cherry-MAD results in the strong recruitment of cherry-MAD to the plasma membrane (*Figure 7E*; *Figure 7—*

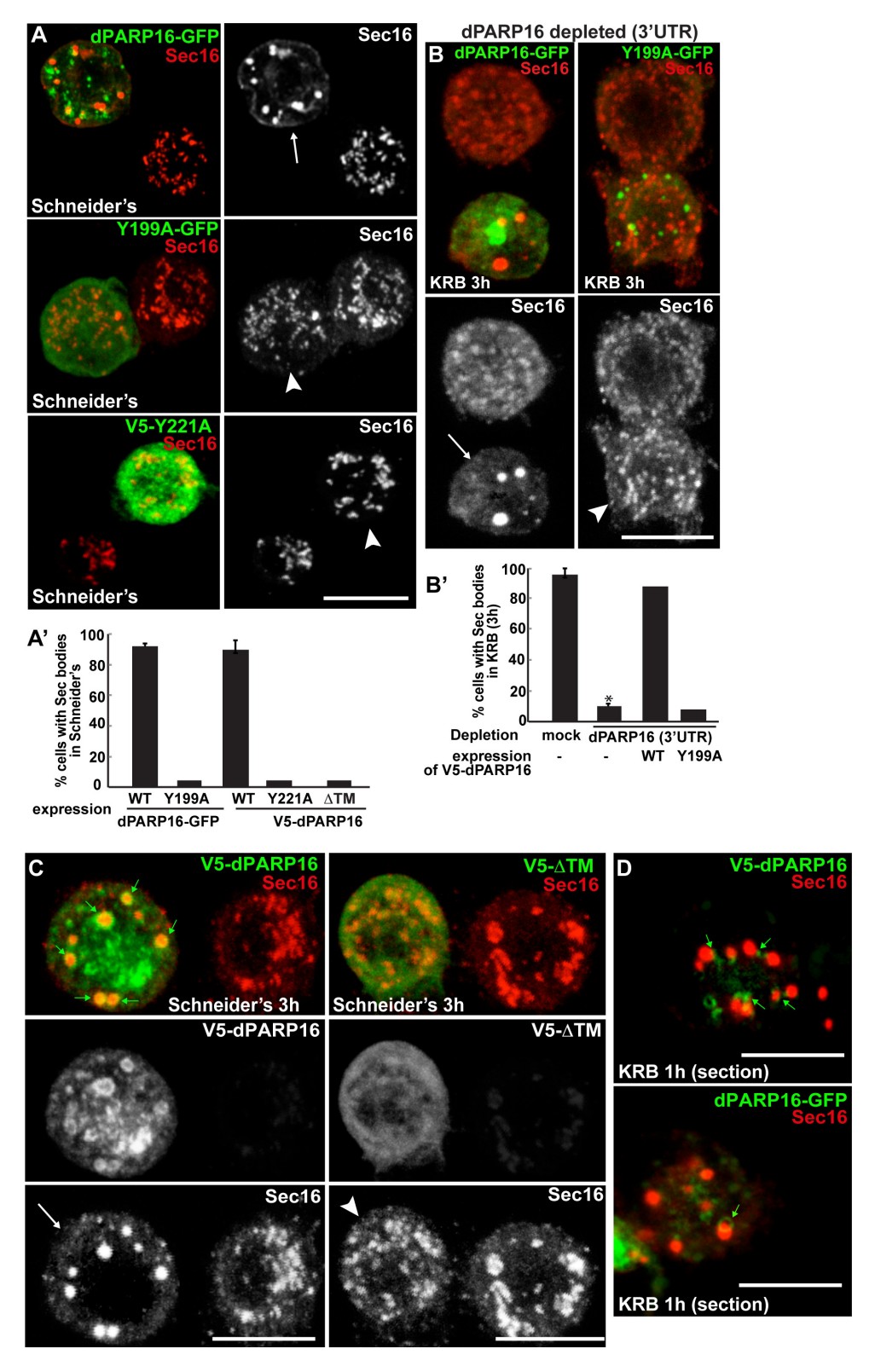

**Figure 5.** dPARP16 catalytic activity and membrane anchoring is required for Sec body formation. (**A, A'**) IF visualisation of Sec body formation (Sec16, red) in growing S2 cells (Schneider's) expressing wild type and catalytic mutant dPARP16 (Y199A and Y221A) (green) (**B**). Note that the expression of the catalytic mutants does not drive Sec body formation (arrowhead in **B**) whereas the wild type dPARP16 does (arrow in **A**) (quantified in **A'**). (**B, B'**) IF visualisation of Sec body formation (Sec16, red) in amino-acid starved S2 cells depleted of dPARP16 (3'UTR) and expressing wild type dPARP16 and

*Figure 5 continued on next page*

*Figure 5 continued*

Y199A dPARP16 catalytic mutant (**B**). Note that the mutant does not rescue Sec body formation (arrowhead in **C**) whereas the wild type dPARP16 does (arrow in **B**) (quantified in **B′**). (**C**) IF visualisation of Sec16 (red) upon wild type V5-dPARP16 and ΔTM V5-dPARP16 expression (green) for 3 hr in Schneider's. Note that overexpressed wild type dPARP16 forms rings and spots (green arrows) as well as Sec bodies (white arrows) whereas the ΔTM V5 dPARP16 does not (arrowheads) (quantified in **A′**). (**D**) IF visualisation in confocal sections of V5-dPARP16 and dPARP16-GFP (green) and Sec16 (red) after 1 hr incubation in KRB. Note that the forming Sec bodies localise closely to dPARP16. Scale bars: 10 μm. Error bars: SEM.

*figure supplement 2B*). Taken together, these results show that upon amino-acid starvation, cherry-MAD binds the MARylated C-terminus of Sec16.

To narrow down the MARylated sequence of the Sec16 C-terminus, we focused on a region of 140 amino-acids that we previously identified as required for the response to serum starvation ('Starvation Response Domain', SRD, 1740–1880) (*Zacharogianni et al., 2011*) (*Figure 7E*). However, expression of SRD-GFP-CAAX in amino-acid starved cells does not lead to the recruitment of cherry-MAD to the plasma membrane (*Figure 7E*; *Figure 7—figure supplement 2C*).

Upon comparison of the SRD sequence in all eukaryotes, we noticed a conserved sequence of 44 amino-acids (1805–1848), that we called SRDC (*Figure 7F*). Strikingly, expression of SRDC-GFP-CAAX leads to the robust recruitment of cherry-MAD to the plasma membrane upon amino-acid starvation (*Figure 7E*; *Figure 7—figure supplement 2D*). In agreement, Sec16 lacking SRDC (ΔSRDC-GFP-CAAX) in Sec16 depleted cells (to prevent oligomerisation with endogenous Sec16) is unable to recruit cherry-MAD to the plasma membrane (*Figure 7E*; *Figure 7—figure supplement 2E*). This shows that this sequence is MARylated upon amino-acid starvation.

To begin to show the functionality of SRDC in Sec body formation, we expressed SRDC-GFP in growing Drosophila cells. Strikingly, this results in the efficient formation of the Sec bodies in the absence of stress (*Figure 8A,D*), a phenotype strongly reminiscent of dPARP16 overexpression (*Figure 4D*). Accordingly, we found that dPARP16 is critically necessary for SRDC-induced Sec body formation, since they were not formed in dPARP16 depleted cells (*Figure 8B,D*). Importantly, SRD-GFP expression does not induced Sec body formation (*Figure 8C,D*). We propose that SRDC MARylation by dPARP16 is one of the events that initiate/drive Sec body formation.

To confirm this, we performed a rescue experiment in cells depleted of Sec16. As expected, Sec16 depleted cells do not form Sec bodies (here marked by Sec23) upon amino-acid starvation (*Figure 8H*), and transfection of Sec16-GFP (*Figure 8E,H*) and SRDC-GFP (*Figure 8F,H*) significantly rescues this formation. However, transfection of ΔSRDC-GFP does not, demonstrating the direct role of SRDC in Sec body formation.

Taken together, using the MAD probe, we show a strong MARylation response upon amino-acid starvation, as Sec16 SRDC is MARylated in a dPARP16 dependent manner, a key event that initiates Sec body formation.

## Discussion

### MAD, a specific probe to detect MARylation in vivo and in vitro

By using biological modules known as macrodomains that do not posses any hydrolase activity, we built and optimised a specific and stable Mono-ADP-ribosylation detection probe (MAD) that has important specifications: First, MAD specifically detects and binds Mono-ADP-ribose. This specificity is sustained by three arguments: (i) MAD is designed and built using PARP14 macrodomains 1–3. Crystallography and calorimetric assays show that each macrodomain has a conserved fold with high binding affinity for mono-ADP-ribose (particularly macrodomains 2 and 3) and the affinity required for in vivo visualisation is provided by the three macrodomains together in a cooperative manner. (ii) We show that the mutated G1055E probe (affecting the binding pocket of macrodomain 2 in such as way that it does not bind mono-ADP-ribose any longer) does not elicit a pattern upon amino-acid starvation in vivo. (iii) The amino-acid starvation GFP-MAD pattern is abrogated upon depletion of dPARP16, the closest homologue of an established human MARylation enzyme PARP16. Conversely, overexpression of dPARP16 elicits GFP-MAD spot formation (but not YFP-PAD spots, which rule out PARylation events). This is confirmed by the use of a PARylation detection probe YFP-PAD. It allows

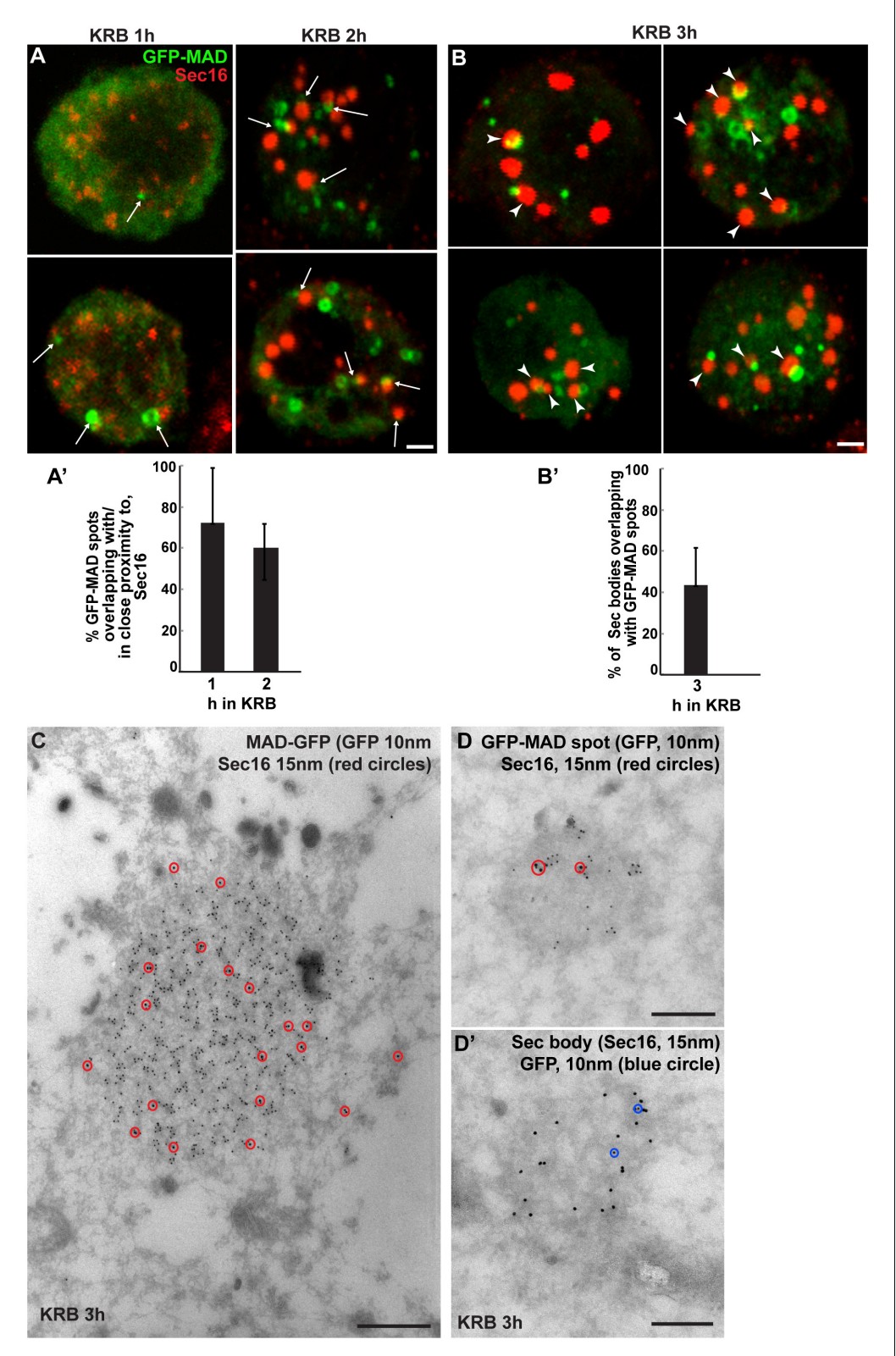

**Figure 6.** Sec bodies are formed in close proximity to GFP-MAD spots. (A, A') Visualisation of GFP-MAD and Sec16 (red) upon 1–2 hr incubation in KRB. The formed GFP-MAD spot are closed to, or overlap with, ERES (1 hr) and small Sec bodies (2 hr) (arrows) (quantified in A'). (B, B') Visualisation of GFP-MAD and Sec16 (red) upon 3 hr incubation in KRB. The forming/formed Sec bodies are adjacent to MAD spots (arrowheads) (quantified in B'). (C) IEM of a GFP-MAD spot (GFP, 10 nm gold) and Sec16 (15 nm) in cells incubated in KRB for 3 hr. Note that a small pool of Sec16 is present within the

*Figure 6 continued on next page*

Figure 6 continued

GFP-MAD spots. (**D–D'**) IEM of GFP-MAD (10 nm gold) and Sec16 (15 nm). A small fraction of Sec16 is found in a GFP-MAD spot (**D**) and conversely, a small fraction of GFP-MAD is present in Sec bodies (both of the presented structures are found in the same cell). Scale bars: 1 μm (**A,B**) and 200 nm (**C–D'**). Error bars: SEM.

the visualization of PARylation events consistent with those reported to occur to RNA binding proteins upon arsenate treatment leading to stress granule formation (*Leung et al., 2011*; *Gagné et al., 2008*). However, YFP-PAD is not remodelled during amino-acid starvation suggesting that PARylation is not prominent during this stress. Taken together, these evidences demonstrate the specificity of GFP-MAD in binding mono-ADP-ribose and detecting MARylation events.

Second, MAD can be used in cells to follow MARylation in real time. In this regard, GFP-MAD allows the visualisation of an unprecedented display of MARylation events upon amino-acid starvation (but not heat shock or arsenate poisoning) under the form of non-membrane bound, reversible spots/rings in the cytoplasm. To the best of our knowledge, this is the first time that MARylation response is visualised in real time during stress.

Third, GFP-MAD has also allowed us to identify dPARP16 as the enzyme catalysing these events in defined regions of the cytoplasm in a dynamic manner. We propose that GFP-MAD spots represent a concentration of MARylated substrates reflecting local dPARP16 activation. This is supported by their co-localization upon expression of both enzyme and probe. This makes GFP-MAD, an efficient activity sensor of the nutrient stress response and dPARP16 the key MARylation enzyme eliciting this response.

Fourth, GFP-MAD can be used for in vitro approaches such as IP and this lead us to the identification of Sec16 as a MARylation substrate. Last, GFP-MAD and cherry-MAD can be used in vivo and in an anchor-away MARylation assay as an alternative to in vitro approaches using purified components. This has allowed us to visualized MARylation in real time and map the MARylated region of Sec16.

## dPARP16, Sec16 and Sec body formation

dPARP16 is necessary and sufficient for Sec body formation upon amino-acid starvation and this creates an unprecedented link between MARylation, metabolic stress and the formation of stress assembly. dPARP16 is necessary for cell survival during amino-acid starvation and recovery. This makes dPARP16 a key enzyme for the cells to specifically cope with amino-acid starvation, as the viability of dPARP16 depleted cells kept in full medium is not compromised. This makes dPARP16 a key survival factor upon amino-acid starvation.

Nutrient starvation in yeast also leads to storage of metabolic enzymes in reversible assemblies (*Narayanaswamy et al., 2009*), such as glutamine synthetase (*Petrovska et al., 2014*) or proteasome subunits (*Peters et al., 2013*). Although no PARPs have been identified in Saccharomyces cerevisiae, the regulation of their organisation might be controlled by SIRT, another class of NAD+ dependent protein that also display ADP-ribosylation activity (*Bütepage et al., 2015*). Conversely, given the abundance of PARPs with predicted MARylation activity in the mammalian genome, it is likely that additional ones, will be required and/or involved in the formation of stress assemblies upon different biological processes, including metabolic stress as described here. We have reported that large Sec bodies did not form in mammalian cells upon conditions used for Drosophila cells, although a remodelling of the early secretory pathway was observed (*Zacharogianni et al., 2014*). Therefore, the fine dissection of the signalling pathways involved in Sec body formation will allow us to recapitulate conditions to trigger their formation in mammalian cells and tissues.

According to our RNAseq data of S2 cells in growing conditions and upon amino-acid starvation conditions (unpublished), dPARP16 has a very low transcriptional level when compared to most genes, suggesting that its protein level is also low. Because dPARP16 moderate overexpression in growing cells leads the detection of MARylation events, it suggests that dPARP16 overexpression leads to its activation. This also suggests that dPARP16 level needs to be kept low in basal conditions to avoid its activation, challenging the detection of its activity in basal conditions. Conversely, dPARP16 is essential during stress, at least amino-acid starvation as depletion of dPARP16 affects the viability of cells during the stress period. How is PARP16 activated upon amino-acid starvation

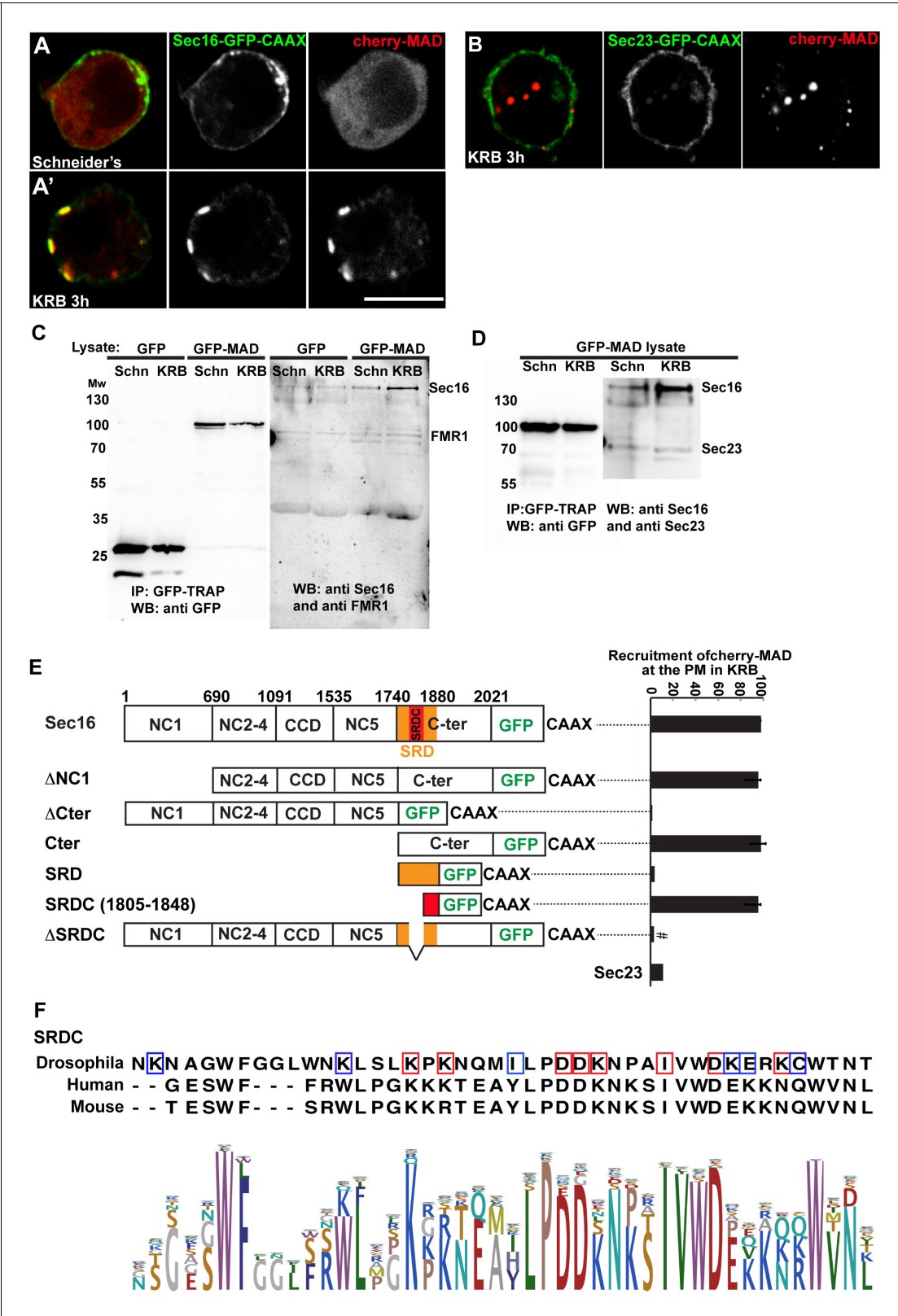

**Figure 7.** Sec16 SRCD is MARylated. (**A, A'**) Co-visualisation of full length Sec16-GFP-CAAX and Cherry-MAD in growing (Schneider's) S2 cells (**A**) and upon amino-acid starvation (KRB) (**A'**). Note that Sec16-GFP-CAAX localises to the plasma membrane where cherry-MAD is recruited upon amino-acid starvation (KRB), whereas in Schneider's, it remains cytoplasmic. (**B**) Co-visualisation of full length Sec23-GFP-CAAX (**B**) and cherry-MAD upon amino-acid starvation (KRB). Note that cherry-MAD is not recruited to the plasma membrane and forms spots in the cytoplasm. (**C**) WB (using anti GFP, anti

*Figure 7 continued on next page*

*Figure 7 continued*

Sec16, anti FMR1) of GFP-MAD immuno-precipitation (IP) using GFP-TRAP from stable S2 cell lines expressing GFP-MAD and GFP, either in growing conditions (Schneider's) or upon amino-acid starvation (KRB 3 hr). (D) WB (using anti GFP, anti Sec16, anti Sec23) of GFP-MAD IP from stable S2 cells expressing GFP-MAD, either in growing conditions (Schneider's) or upon amino-acid starvation (KRB 3 hr). Note that Sec23 pull-down by GFP-MAD is very weak when compared to Sec16. (E) Map of all Sec16-GFP-CAAX truncations used and the quantitation of cherry-MAD recruitment to the plasma membrane. Note that Sec16-ΔSRDC-GFP-CAAX transfection was performed in Sec16 depleted cells (marked by #) to avoid oligomerisation with endogenous Sec16. (F) Muscle sequence alignment of Drosophila, human and mouse Sec16 SRDC (1805–1848) and presentation of logo sequence outlining the degree of sequence conservation among all eukaryotes (defined using CLC Main Workbench 6.7.1 using the full length Sec16 sequence against all eukaryote sequences from the non-redundant protein database using standard settings). The red squares indicate the conserved residues that are potentially MARylated and the blue squares the non-conserved ones.

The following figure supplements are available for figure 7:

**Figure supplement 1.** Sec16-CAAX recruits Sec23 to the PM upon AA starvation.

**Figure supplement 2.** Sec16 SRDC is MARylated upon amino-acid starvation.

remains to be elucidated. Given the nature of the stress, TORC1 activation would be an ideal pathway but we have shown that it is not involved in Sec body formation (*Zacharogianni et al., 2014*). Another possibility is the fluctuation in the intracellular pH as shown for yeast upon energy deprivation that results in the formation of macromolecular assemblies (*Munder et al., 2016*). However, the time scale is very different (minutes for the drop in pH fluctuation versus hours of starvation for Sec body formation). Furthermore, the assemblies that form upon pH fluctuation do not appear to be mediated by a signalling pathway. This remains to be investigated.

In mammalian cells, PARP16 is activated via auto-MARylation triggered by ER stress (*Jwa and Chang, 2012*). As a result, it MARylates two key kinases of the ER stress response UPR, Ire1 and PERK (*Gardner et al., 2013*), leading to their activation (*Jwa and Chang, 2012*) and the unfolded Protein Response. As dPARP16 shares many features with its human counterpart, ER stress could in principle lead to Sec body formation. However, we have previously shown (*Zacharogianni et al., 2014*) and our unpublished results) that inducing ER stress does not lead to Sec body formation. Although ER stress might be involved in the amino-acid starvation stress response, it is not sufficient to trigger it. This result is reinforced by the demonstration that cell survival is significantly more affected by amino-acid starvation than by ER stress (*Figure 8—figure supplement 1*). As a result, dPARP16 appears to be more critical for amino-acid starvation than for ER stress (*Figure 8—figure supplement 1*).

This suggests that additional signals are generated during amino-acid starvation. These are under investigation. Another substrate of mammalian PARP16 is karyopherin (*Di Paola et al., 2012*), a component required for nuclear export. However, our unpublished results shows that the pharmacological inhibition of nuclear export does not inhibit Sec body formation (not shown), suggesting that at least during amino-acid starvation, karyopherin might not play a prominent role and that dPARP16 has different substrates.

One of these substrates is the ERES/Sec body component Sec16 and more specifically a conserved 44 amino-acid sequence (SRDC) in its C-terminus. Indeed, The CAAX version of SRDC recruits cherry MAD to the plasma membrane. Overexpression of SRDC leads to the formation of Sec bodies in a dPARP16 dependent manner, and SRCD rescues Sec body formation in Sec16 depleted cells. This suggests that Sec16-SRDC MARylation is a triggering event in Sec body formation.

The discovery of a short peptide required for the formation of Sec bodies is reminiscent to the existence of an Amyloid Converting Peptide in proteins found in nuclear amyloid bodies in cells upon several stress (*Audas et al., 2016*). Interestingly, amyloidogenesis is mediated by this motif binding a long non-coding RNA that could be equivalent or comparable to the SRDC MARylation during Sec body formation. Whether the SRCD sequence is also also present in other proteins recruited to Sec bodies remains to be investigated.

In the context of the full-length endogenous protein, SRDC MARylation could act as both a signalling and structural event allowing the recruitment of Sec16 and other ERES components into Sec bodies. However, SRCD on its own is not recruited to Sec bodies, and our interpretation is that it

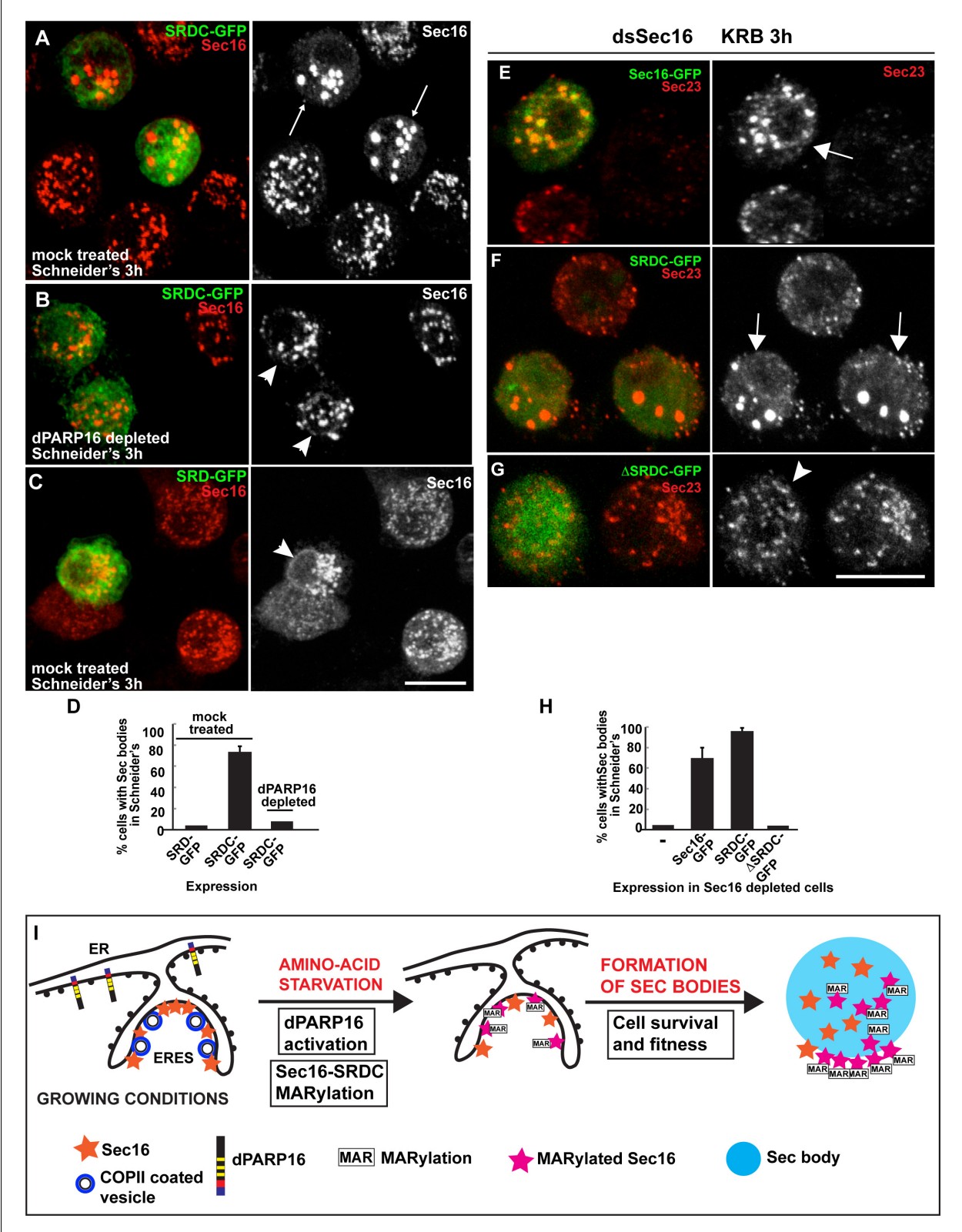

**Figure 8.** Sec16 SRDC MARylation is critical for Sec body formation. (**A–D**) Visualisation of endogenous Sec16 (red) in mock treated (**A,C**) and dPARP16 depleted S2 cells (**B**) in Schneider's transfected with the Sec domains SRDC-GFP (**A,C**) and SRD-GFP (**B**). Note that expression of SRDC leads to the formation of Sec bodies, similar to dPARP16 overexpression and that dPARP16 depletion prevents this formation (quantified in D). ( **E–H**) Visualisation of endogenous Sec23 (to mark Sec bodies) in starved Sec16 depleted cells (ds Sec16) transfected with Sec16-GFP (**E**), SRDC-GFP (**F**) and ΔSRDC-GFP

*Figure 8 continued on next page*

*Figure 8 continued*

(**G**). Note that Sec16 depletion prevent cells to form Sec bodies. Both, full length Sec16 and SRDC transfection rescue Sec body formation, but not Sec16 lacking SRDC. Arrows point to cells where Sec bodies have formed and arrowheads where they haven't. (**I**) Model of Sec body formation upon amino-acid starvation of Drosophila S2 cells. Under growing conditions, Sec16 is at ERES where COPII coated vesicles bud. dPARP16 is at the ER. Upon amino-acid starvation, dPARP16 is activated and MARylates SRDC, a 44 amino-acid stretch in the C-terminus of Sec16. Sec16 MARylation (and possibly MARylation of other components) triggers the formation of Sec bodies by dynamic incorporation of MARylated substrates within the structure and promotes cell survival and fitness. Scale bars: 10 μm. Error bars: SEM.

The following figure supplement is available for figure 8:

**Figure supplement 1.** Comparison of cell viability upon amino-acid starvation and ER stress.

may act only as a signalling event for Sec body formation. The nature of this signalling need to be further investigated. This is in line with the fact that GFP-MAD is not readily observed within the Sec body core. GFP-MAD appears as a ring/spot at the base of Sec bodies. Although this is consistent with the protein packing and competing binding that takes place during the formation of stress assemblies (that most likely would exclude GFP-MAD), it might also suggest that MARylated Sec16 forms a signalling platform. This would lead to the modifications of other Sec body components allowing their incorporation (*Figure 8I*).

Interestingly, PARylation has been proposed to preferentially occur on LCSs (Low complexity sequences, that is, region of poor amino-acid diversity) (*Leung, 2014*). These are normally thought to correspond to disordered regions. Sec16 is rich in LCSs (*Zacharogianni et al., 2014*) and SRDC is intrinsically disordered, therefore accessible to be modified by dPARP16. Taken together, we postulate that Sec16 is a stress response protein and a new substrate for the pro-survival dPARP16 upon amino-acid starvation

## Materials and methods

### Cell culture, amino acid starvation, depletions (RNAi) and transfections

Drosophila S2 cells with a non authenticated identity (the original source is unknown but they have been used in the lab for the last 15 years and they are mycoplasma free) were cultured in Schneider's medium (Sigma) supplemented with 10% insect tested foetal bovine serum at 26°C as described in (*Kondylis and Rabouille, 2003*; *Kondylis et al., 2007*). Amino acid starvation of cells for 3 or 4 hr was performed using Krebs Ringer's Bicarbonate buffer (10 mM D(+) Glucose; 0.5 mM $MgCl_2$; 4.5 mM KCl; 121 mM NaCl; 0.7 mM $Na_2HPO_4$; 1.5 mM $NaH_2PO_4$ and 15 mM sodium bicarbonate) at pH 7.4 (*Zacharogianni et al., 2014*)

Wild type Drosophila S2 cells were depleted by dsRNAi, as previously described (*Kondylis and Rabouille, 2003*; *Kondylis et al., 2007*). Cells were analysed after incubation with dsRNAs for five days typically leading to depletion in more than 90% of the cells.

Transient transfections of PMT constructs (see below) were performed using Effectene transfection reagent (301425; Qiagen, Germany) according to manufactures instructions. Expression was induced 48 hr after transfection with 1 mM $CuSO_4$ for 1.5 hr (*Zacharogianni and Rabouille, 2013*).

Stable cell lines expressing GFP-MAD and GFP are maintained in Schneider's supplemented medium with 300ug/ml Hygromycin B. Plasmid expression is induced with 1 mM $CuSO_4$ for 1-2 hr.

### Antibodies

The following antibodies were used: Rabbit polyclonal anti-Sec16 (*Ivan et al., 2008*)1:800 IF, 1:2500 WB; Rabbit polyclonal anti-Sec23 (RRID:AB_2546460, Thermo scientific, 1:200 IF, 1:500 WB); Mouse monoclonal anti-V5 (ThermoFischer Scientific 46–0705, 1:500 IF); Rabbit polyclonal anti-V5 (RRID: AB_261889, Sigma V8137); Mouse monoclonal anti-ATP5A (RRID:AB_301447, Abcam 15H4C4; 1:1000 IF); Mouse monoclonal anti-KDEL receptor (RRID:AB_1209241; Abcam ab69659, 1:500 IF); Mouse monoclonal anti-calnexin 99A (Gift fom Sean Munro, 1:10 IF); Mouse monoclonal anti-FMR1 (RRID:AB_528251, DSHB supernatant clone 5A11, 1:800 IF, 1:2000 WB); Rabbit polyclonal anti-GFP (RRID:AB_1002458, Acris antibodies, 1: 5000 WB); Rabbit polyclonal anti-GFP 1:100 IEM; Polyclonal

FMR1-c 1:20 IF 1:500 WB (DSHB), anti-Rabbit HRP (RRID:AB_384736, GE healthcare 1:2000 IEM); Anti-Mouse HRP (RRID:AB_384734, GE Healthcare, 1:2000 IEM).

## PMT-DNA constructs and dsRNAs

All the primers used for generating the DNA constructs and RNAi probes are listed in *Figure 1— source data 1*. To generate the pMT-sfGFP vector, super folder (sf) GFP was amplified and cloned into pMT-V5 using *SacII* and *PmeI* restriction sites replacing the V5 tag with sfGFP.

The sequence corresponding to the ORFs of CG40441 (dARTD1/PARP1), CG4719(dARTD5-6/ dTankyrase) and CG15925(dARTD15/dPARP16) were amplified from a cDNA library made from Drosophila S2 cells and clone into pMT-sfGFP using *KpnI* and *ApaI*.

To generate pMT-V5-dPARP16, dPARP16 was amplified from dPARP16-GFP and cloned into pMT-V5 using *AgeI* and *PmeI*. To generate the mutant pMT-Y199A-dPARP16-GFP, dPARP16 was amplified using primers harbouring a mutation at position Y199A and cloned into pMT-GFP using *KpnI* and *ApaI*.

To generate the mutant PMT-V5 Y221A-dPARP16, dPARP16 was amplified using primers harbouring a mutation at position Y221A and cloned into pMT-V5 using *AgeI* and *PmeI*.

To generate the truncated pMT-ΔTM-V5-dPARP16, dPARP16 was amplified and cloned into pMT-V5 using *AgeI* and *PmeI*.

To generate the pMT-CAAX-sfGFP vector, the sequence corresponding to C-terminus CAAX motif of Ras (SGLRSRAQASNSRVKMSKDGKKKKKKSKTKCVIM) was amplified and cloned into pMT-sfGFP using *AgeI* and *PmeI*. The Sec16 truncations: ΔNC1, ΔCter; Cter, SRD and SRDC were cloned into pMT-CAAX-sfGFP using *EcoI* and *ApaI*.

To generate the pMT-Sec16ΔSRDC-sfGFP and the pMT-CAAX-Sec16-ΔSRDC-sfGFP the SRDC deleted version was cross amplified using fusion primers (*Figure 1—source data 1*) and cloned into pMT-Sec16Fl-sfGFP and pMT-CAAX-Sec16Fl-sfGFP respectively using *EcoRvI* and *SacII*

The dsRNAs used for RNAi of dPARP1, dTNK and dPARP16 were amplified using primers harbouring T7 promoters in their sequence and used for in vitro transcription using the T7 Megascript Kit (AMBION) to generate the dsRNAs.

To generate GFP-MAD, the macrodomains 1–3 of human PARP14 were amplified from cDNA of human HEK293 cells and cloned into pMT-GFP using *AgeI* and *PmeI* followed by the insertion of a Hex-HIS-TEV-linker using *AgeI*. To generate the GFP-MAD-Macro2 mutant, the macrodomains 1–3 of MAD were amplified using primers harbouring the G1055E mutation followed by the insertion of a Hex-HIS-TEV linker as described above.

To generate YFP-PAD, YFP was amplified from a YFP-plasmid and cloned into pMT-sfGFP with *AgeI* and *ApaI* replacing sf-GFP with SYFP. H2A1.1 was amplified from a pUCIDT plasmid synthesized by (IDT) and cloned into pMT-SYFP with *AgeI* and *PmeI*, followed by the insertion of a Hex-HIS-TEV linker as described above.

## Immunofluorescence (IF)

Drosophila S2 cells were plated on glass coverslips, treated as described, fixed in 4% PFA in PBS for 20 min and processed for inmunofluorescence as previously described (*Kondylis and Rabouille, 2003*; *Zacharogianni and Rabouille, 2013*). Samples were viewed under a Leica SPE confocal microscope using a 63x oil lens and 2-4x zoom. 14 to 20 planes were projected to capture the whole cell that is displayed unless indicated otherwise.

## Immuno-electron microscopy (IEM) and correlative GFP fluorescence/ IEM

IEM of dPARP16 was performed as described previously (*Kondylis et al., 2007*; *van Donselaar et al., 2007*). The correlative Fluorescence/IEM method (*Hassink et al., 2012*) is adapted from (*Vicidomini et al., 2010*). Briefly, S2 cells stably expressing GFP-MAD were incubated in KRB for 1 and 3 hr, fixed with 4% PFA (in 0.1M PB) for 3 hr followed by 1% PFA overnight. Ultrathin sections were cut, picked up on electron microscopy copper formvar coated grids, labelled with a goat anti-GFP antibody coupled to biotin followed by a rabbit anti-biotin antibody and ProteinA Gold (10 nm), followed or not by labeling with a rabbit anti Sec16 antibody followed by proteinA Gold 15 nm.

Sections were visualized on a Delta vision fluorescence microscope to detect the fluorescence signal corresponding to GFP. Cell profiles were recorded. The same grid was then viewed in the electron microscope (Jeol) and the ROI was photographed.

### Live imaging experiments

Live imaging of GFP-MAD was performed using S2 cells stably expressing GFP-MAD at 26°C in Schneider's medium (t = 0) and incubated in KRB up to 3 hr. Cells were filmed using a Leica SPE confocal microscope using a 63x lens at 4x zoom. 10 z-planes with a z-step of 0.5 μm were recorded every 10 min.

### Immuno-precipitation and Western blot

$200 \times 10^6$ and $150 \times 10^6$ S2 cells stably expressing GFP-MAD and GFP were incubated for 3 hr at 26°C in KRB and in Schneider's, respectively. Cells were harvested, placed immediately on ice and washed with ice cold PBS by mild centrifugation (1100 rpm, 4 min at 4°C). Cells were lysed in 600 μl lysis buffer (10% glycerol; 1% Triton X100; 50 mM Tris-HCl pH7.5; 150 mM NaCl; 50 mM NaF; 25 mM $Na_2gP$; 1 mM $Na_2VO_3$; 5 mM EDTA and one tablet Roche protease inhibitor/100 ml) for 30 min upon rotation at 4°C. The cell lysate was then centrifuged at 14,000 rpm for 20 min at 4°C. Protein concentration was determined by using BCA protein assay. The cell lysate was added to 20 μl GFP-Trap (R) beads (Chromotek) washed in lysis buffer and incubated by rotation at 4°C. The GFP-Trap beads were then washed 3x for 5 min at 4°C with 1 ml lysis buffer (at 2000 rpm, 2 min at 4°C). The supernatant was collected and boiled for 5 min in 50 μl 2xsample buffer with DTT. Samples (15 mg of protein) were fractionated on a 10% SDS-PAGE gel, proteins transferred to a nitrocellulose membrane. Blotting was done in blocking buffer (PBST with milk), after which the antibodies were added in the concentrations as described above.

### Heat stress and Arsenate treatment

Heat stress was performed on $2 \times 10^6$ Drosophila S2 cells in 3 cm dish in a oven at 37°C (Thermo Electron) for 3 hr as described in (*Jevtov et al., 2015*). Treatment with 0.5 mM $NaAsO_2$ was performed at 26°C for 3 hr.

## Cell survival and fitness upon and after amino-acid starvation and ER stress

0.75 million cells were mock- (dsGFP) and dPARP16 depleted. After five days of depletion the cells proliferated to reach 3.0 million respectively. This was set at 100% (t = 0). For the treatments, cells were either kept in Schneider's, or amino-acid starved in KRB for 3 hr or treated with Schneider's supplemented with 2.0 mM DTT for 3 hr (ER stress). The cells were then washed, and the medium changed to Schneider's allowing recovery for up to 16 hr.

Cell viability was determined by exclusion of Trypan Blue. For each time point, 0.1 ml of cell suspension was mixed with 0.1 ml of 0.4% Trypan. The number of living cells that were counted using a hemocytometer. The cell number was monitored and expressed as a percentage of t = 0.

Experiments were performed in at least three biological replicates each consisting of three technical replications. All technical replicates were averaged.

The error bars in the graphs are standard deviation (SD) calculated over all biological replicates. p-values are indicated in the legend of the figures.

### Quantification and statistics

Two/three biological replicates were performed per experiment. For IF of depleted or treated cells, at least four fields per experiment were analysed comprising at least 50 cells. For transfected cells, at least 30 cells were analysed. Results are expressed as standard deviations.

## Acknowledgements

We thank the Rabouille's lab members and Fulvio Reggiori, Adam Grieve and Tim Levine for critically reading the manuscript as well as Geert Kops and Puck Knipscheer for helpful comments. The work is supported by NWO grant to CR (822-020-016). We thank Anko de Graaff and the Hubrecht Imaging Center for supporting the imaging.

# Additional information

## Funding

| Funder | Grant reference number | Author |
|---|---|---|
| Netherlands Wetenschappe-lijke Organisatie | 822-020-016 | Catherine Rabouille |
| Hubrecht Institute | | Catherine Rabouille |

The funders had no role in study design, data collection and interpretation, or the decision to submit the work for publication.

## Author contributions

AA-G, Designed and executed the experiments, Interpreted the results, Together with CR wrote the manuscript, Conception and design, Acquisition of data, Analysis and interpretation of data, Drafting or revising the article; MMvO, Contributed in making the cell lines expressing the probes and performed the IP, Conception and design, Acquisition of data, Analysis and interpretation of data; TV, Performed the EM related to this paper including the correlative fluorescence/EM, Conception and design, Acquisition of data, Analysis and interpretation of data; CR, Supervised the project, Helped with the design of the experiments and interpretation of the results and wrote the ms with AAG, Conception and design, Acquisition of data, Analysis and interpretation of data, Drafting or revising the article

## Author ORCIDs

Catherine Rabouille, http://orcid.org/0000-0002-3663-9717

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
