## [Decision Letter]

Thank you for submitting your article "In vivo vizualisation of Mono-ADP-ribosylation by dPARP16 upon amino-acid starvation" for consideration by *eLife*. Your article has been reviewed by three peer reviewers, including Benjamin S Glick (Reviewer #2) and a member of our Board of Reviewing Editors, and the evaluation has been overseen Vivek Malhotra as the Senior Editor.

The reviewers have discussed the reviews with one another and the Reviewing Editor has drafted this decision to help you prepare a revised submission. As you can see, the reviewers only have a few suggestions to improve the paper, which mainly have to do with how you lay out the figures and some explanation.

Aguilera et al. use specifically engineered probes to visualize mono and poly ADP-ribosylation (MAR and PARylation) in living cells. The authors find that in insect cells MARylation is triggered by amino acid starvation. MARylation in these starved cells further leads to the formation of Sec bodies, a stress inducible protein assembly that protect cells from the negative consequences of amino acid depletion. Overall this is an interesting and well executed paper. It adds significantly to our understanding of stress response pathways, and to the function of mono-ADP ribosylation, and of Parp16 in particular. It shows for the first time a link between amino acid metabolism, protein ADP-ribosylation and stress-inducible assemblies.

Essential revisions

One issue raised was the following: Is dPARP16 upregulated or activated upon amino acid starvation? Otherwise, as pointed out in the Discussion, if dPAPR16 were constitutively expressed and active, Sec bodies would form under growing conditions. This issue should be addressed if possible in the experimental section.

A second point was Figure 4. The cell survival curve strikes this reviewer as very important, more so than several of the molecular details reported in other figures, yet it was very much under-emphasized in the Introduction, Results and Discussion. The role of Parp16 in protecting cells from stress as measured by cell survival should probably be a stand-alone figure, with appropriate methods and discussion, and perhaps a bit more investigation. From a general stress-response perspective knowing if a pathway is important for survival is the most significant point of all. The reviewer was frustrated that the methods used to generate Figure 4 did not seem to be in the paper, and that the error bars look inappropriate (they may be SEM values which are usually not the right way to quantify error in biological experiments). It was also a surprise that the cells die so quickly in what seems like a relatively mild stress, and wondered if this deserved comment. We would like to see the same curve for a classic ER stressor like tunicamycin or thapsigargin +/- dParp16, since the homolog was implicated in survival in response to these stresses in human cells.

Another concern is with the error bars on Figure 4. They are probably SEM within one experiment, but that isn't a realistic measure of likely overall error. Better to just include all the data, the SD, or some other measure of variability.

---

## [Author Response]

Essential revisions

*One issue raised was the following: Is dPARP16 upregulated or activated upon amino acid starvation? Otherwise, as pointed out in the Discussion, if dPAPR16 were constitutively expressed and active, Sec bodies would form under growing conditions. This issue should be addressed if possible in the experimental section.*

We thank the reviewers for this suggestion and wish to point out that this experiment has been performed in the original manuscript. Indeed dPARP16 overexpression leads to the formation of Sec bodies in growing conditions (shown in Figure 4’ and E). We therefore concluded that dPARP16 is not only necessary but also sufficient for Sec body formation.

We hypothesized that dPARP16 overexpression leads to an increased concentration resulting in its activation via dimerization or oligomerization, as it is the case for many plasma membrane receptors. As we pointed out in the discussion, we propose that in basal condition dPARP16 transcription levels are kept very low to ensure that it remains inactive.

However, it is currently unknown how endogenous PARP16 is activated upon starvation. We have commented on this aspect in the Discussion section.

*A second point was Figure 4. The cell survival curve strikes this reviewer as very important, more so than several of the molecular details reported in other figures, yet it was very much under-emphasized in the Introduction, Results and Discussion. The role of Parp16 in protecting cells from stress as measured by cell survival should probably be a stand-alone figure, with appropriate methods and discussion, and perhaps a bit more investigation. From a general stress-response perspective knowing if a pathway is important for survival is the most significant point of all. The reviewer was frustrated that the methods used to generate Figure 4 did not seem to be in the paper, and that the error bars look inappropriate (they may be SEM values which are usually not the right way to quantify error in biological experiments). It was also a surprise that the cells die so quickly in what seems like a relatively mild stress, and wondered if this deserved comment. We would like to see the same curve for a classic ER stressor like tunicamycin or thapsigargin +/- dParp16, since the homolog was implicated in survival in response to these stresses in human cells.*

As suggested by the reviewer, we have repeated the experiment monitoring the role of dPARP16 in cell survival to amino-acid starvation. As requested, we have also included a classical ER stress (using DTT at 2mM), a concentration known in the field to trigger this stress readily.

The results are presented in Figure 4 and Figure 8—figure supplement 1 (as a standalone).

In Figure 4, we confirm that upon amino-acid starvation in dPARP16 depleted cells and that the cell survival is significantly lower compared to the mock depleted. Furthermore, upon stress relief, the cell recovery in dPARP16 depleted cells is significantly lower compared to the mock depleted. Importantly, the cell survival is unaffected in dPARP16 depleted cells that are maintained in growing conditions. Given the role of dPARP16 in Sec body formation (this manuscript) and given the role of Sec bodies in cell survival (Zacharoggianni et al. 2014), we propose that dPARP16 critically helps to protect the cells from amino-acid starvation through its crucial role in the formation of the pro-survival Sec bodies.

Taken together, this allows us to postulate dPARP16 as a key survival factor during amino-acid starvation. This has been now clearly stated in the Introduction, Results and Discussion.

In Figure 8—figure supplement 1, we included the ER stress results thus allowing the comparison between amino-acid starvation and DTT treatment. DTT treatment for 3h only leads to a slight decrease in cell viability when compared to amino-acid starvation. This clearly demonstrates that amino-acid starvation is not a middle stress and is in fact stronger than ER stress (and heat shock as shown in Jevtov et al., 2015). Furthermore, dPARP16 depletion has hardly any effect on the survival to DTT treatment, whereas it has a very strong effect on amino-acid starved cells. This shows that in *Drosophila* cells, dPARP16 is needed to respond to the starvation stress.

The reason for which we have separated these two sets of results is that the DTT experiment did not fit in Figure 4 but were best discussed in the context of what is known for human PARP16 in ER stress.

*Another concern is with the error bars on Figure 4. They are probably SEM within one experiment, but that isn't a realistic measure of likely overall error. Better to just include all the data, the SD, or some other measure of variability.*

We now present the results using standard deviation (to display the biological variability, as requested by the reviewer) based on at least 3 biological replicates, each comprising 3 technical replicates that were averaged. We added all the details about the experimental procedure and analysis in the Materials and methods.